# Effects of a Primary Care Antimicrobial Stewardship Program on Meticillin-Resistant *Staphylococcus aureus* Strains across a Region of Catalunya (Spain) over 5 Years

**DOI:** 10.3390/antibiotics13010092

**Published:** 2024-01-18

**Authors:** Alfredo Jover-Sáenz, María Ramírez-Hidalgo, Alba Bellés Bellés, Esther Ribes Murillo, Meritxell Batlle Bosch, Anna Ribé Miró, Alba Mari López, José Cayado Cabanillas, Neus Piqué Palacín, Sònia Garrido-Calvo, Mireia Ortiz Valls, María Isabel Gracia Vilas, Laura Gros Navés, María Jesús Javierre Caudevilla, Lidia Montull Navarro, Cecilia Bañeres Argiles, Pilar Vaqué Castilla, José Javier Ichart Tomás, Mireia Saura Codina, Ester Andreu Mayor, Roser Martorell Solé, Ana Vena Martínez, José Manuel Albalad Samper, Susana Cano Marrón, Cristina Soler Elcacho, Andrés Rodríguez Garrocho, Gemma Terrer Manrique, Antoni Solé Curcó, David de la Rica Escuin, María José Estadella Servalls, Ana M. Figueres Farreny, Luís Miguel Montaña Esteban, Lidia Sanz Borrell, Arancha Morales Valle, Mercè Pallerola Planes, Aly Hamadi, Francesc Pujol Aymerich, Francisca Toribio Redondo, María Cruz Urgelés Castillón, Juan Valgañon Palacios, Marc Olivart Parejo, Joan Torres-Puig-gros

**Affiliations:** 1Unidad Territorial Infección Nosocomial (UTIN), Hospital Universitari Arnau de Vilanova de Lleida (HUAV), 25198 Lleida, Spain; mframirez.lleida.ics@gencat.cat; 2Sección de Microbiología, Hospital Universitari Arnau de Vilanova de Lleida (HUAV), 25198 Lleida, Spain; abelles.lleida.ics@gencat.cat; 3Unidad de Farmacia de Atención Primaria, Institut Català de la Salut (ICS), 25007 Lleida, Spain; eribes.lleida.ics@gencat.cat; 4Equipo de Atención Priamaria (EAP) Les Borges Blanques, 25400 Lleida, Spain; mbatlle.lleida.ics@gencat.cat (M.B.B.); aribem.lleida.ics@gencat.cat (A.R.M.); 5EAP Pla d’Urgell, 25001 Lleida, Spain; amari.lleida.ics@gencat.cat (A.M.L.); jcayado.lleida.ics@gencat.cat (J.C.C.); npique.lleida.ics@gencat.cat (N.P.P.); 6EAP Balàfia-Pardinyes, 25005 Lleida, Spain; sgarrido.lleida.ics@gencat.cat (S.G.-C.); mortiz.lleida.ics@gencat.cat (M.O.V.); 7EAP Rambla de Ferran, 25007 Lleida, Spain; mgracia.lleida.ics@gencat.cat; 8EAP Lleida Rural Nord, 25110 Lleida, Spain; lgros@gss.cat; 9Centre Penitenciari de Ponent, 25199 Lleida, Spain; jjavierrec@gencat.cat; 10EAP Eixample, 25006 Lleida, Spain; lmontull.lleida.ics@gencat.cat (L.M.N.); cbaneres.lleida.ics@gencat.cat (C.B.A.); 11EAP Primer de Maig, 25002 Lleida, Spain; pvaque.lleida.ics@gencat.cat; 12Servicio de Urgencias, Hospital Universitari Arnau de Vilanova de Lleida (HUAV), 25198 Lleida, Spain; jxichart.lleida.ics@gencat.cat (J.J.I.T.); msaura.lleida.ics@gencat.cat (M.S.C.); 13Col·legi Oficial de Podòlegs, 25001 Lleida, Spain; eandreu.lleida.ics@gencat.cat; 14EAP Cervera, 25200 Lleida, Spain; rmartorell.lleida.ics@gencat.cat; 15Servei de Geriatria, Hospital Universitari Santa Maria, 25198 Lleida, Spain; anav@gss.cat; 16EAP Ponts, 25740 Lleida, Spain; jalbalad.lleida.ics@gencat.cat; 17EAP Onze de Setembre, 25005 Lleida, Spain; scano.lleida.ics@gencat.cat (S.C.M.); csoler.lleida.ics@gencat.cat (C.S.E.); arodriguezg.lleida.ics@gencat.cat (A.R.G.); gterrer.lleida.ics@gencat.cat (G.T.M.); 18EAP Bellpuig, 25250 Lleida, Spain; ajsole.lleida.ics@gencat.cat; 19EAP Artesa de Segre, 25730 Lleida, Spain; drica.lleida.ics@gencat.cat; 20EAP Cappont, 25001 Lleida, Spain; mjestadella.lleida.ics@gencat.cat; 21EAP Almacelles, 25100 Lleida, Spain; afigueres.lleida.ics@gencat.cat; 22EAP Seròs, 25183 Lleida, Spain; lmontana.lleida.ics@gencat.cat (L.M.M.E.); lsanz.lleida.ics@gencat.cat (L.S.B.); 23EAP Lleida Rural Sud, 25171 Lleida, Spain; amorales.lleida.ics@gencat.cat; 24EAP Balaguer, 25600 Lleida, Spain; mpallerola.lleida.ics@gencat.cat (M.P.P.); ahamadi.lleida.ics@gencat.cat (A.H.); 25EAP Alcarràs, 25180 Lleida, Spain; fpujol.lleida.ics@gencat.cat; 26EAP Alfarràs-Almenar, 25120 Lleida, Spain; ftoribio.lleida.ics@gencat.cat; 27EAP Bordeta-Magraners, 25001 Lleida, Spain; curgeles.lleida.ics@gencat.cat; 28EAP La Granadella, 25177 Lleida, Spain; jvalganon.lleida.ics@gencat.cat; 29EAP Tàrrega, 25300 Lleida, Spain; molivart.lleida.ics@gencat.cat; 30Departament de Salut Pública, Universitat de Lleida (UdL), 25006 Lleida, Spain; joan.torres1958@gmail.com

**Keywords:** antimicrobial stewardship, use antimicrobials, multidrug-resistant microorganisms, community-onset, epidemiology, MRSA

## Abstract

Primary care antimicrobial stewardship program (ASP) interventions can reduce the over-prescription of unnecessary antibiotics, but the impact on the reduction in bacterial resistance is less known, and there is a lack of available data. We implemented a prolonged educational counseling ASP in a large regional outpatient setting to assess its feasibility and effectiveness. Over a 5-year post-implementation period, which was compared to a pre-intervention period, a significant reduction in antibiotic prescriptions occurred, particularly those associated with greater harmful effects and resistance selection. There was also a decrease in methicillin-resistant *Staphylococcus aureus* (MRSA) strains and in their co-resistance to other antibiotics, particularly those with an ecological impact.

## 1. Introduction

*Staphylococcus aureus* is a microorganism recognized for being both a commensal and an opportunistic pathogen in humans and animals [1]. The methicillin-resistant *Staphylococcus aureus* (MRSA) strain has become a relevant lineage, with a continuously increasing prevalence in hospitals, communities, and livestock environments that poses a threat to public health. Moreover, the high pathogenicity of MRSA, which is attributable to various virulence factors, such as SCCmec acquired through genetic transfer from the mecA gene, as well as antibiotic resistance, compromises host immunity, making it responsible for causing severe infections in both humans and animals [2].

Traditionally, MRSA has been considered one of the primary multidrug-resistant pathogens causing healthcare-associated infections (HA-MRSA), and it has reached endemic proportions in many countries. It has become a leading cause and potentially fatal agent of invasive infections, skin and soft tissue infections, and pneumonia [3]. In the United States, the estimated annual cost of these infections is around USD 2.7 million, with a significant loss of lives that amounts to 20,000 deaths per year [4,5]. Alarmingly, its aggressive nature has extended to the community setting in the last two decades, where it is known as community-acquired MRSA (CA-MRSA), with greater pathogenicity and transmissibility affecting both young and healthy individuals [6]. In this context, the level of colonization in the general population can increase in environments with a high presence of livestock animals, as observed in Catalonia, Spain, where 75.6% of pig industry workers are colonized by MRSA, particularly with the ST398 strain [7].

Strategies to prevent acquisition rely not only on controlling the spread of clones and horizontal gene transfer, but also on reducing antibiotic pressure in the environment. There is a clear association between the volume of antibiotic prescriptions and the presence of multidrug-resistant microorganisms (MDR) [8,9]. This prevalence may be even higher when broad-spectrum antibiotics are used. Recent epidemiological studies conducted in our country showed a consistently high prevalence of MRSA in the community, exceeding 10% over the last decade [10]. Unfortunately, despite this, it is not only the antibiotic prescription rates in Spanish primary care that are high; the level of use of broad-spectrum antimicrobials remains two to three times higher than that observed in most European countries [11].

Several meta-analyses have demonstrated a direct relationship between exposure to certain antimicrobial classes and microbiological resistance [12]. Cephalosporins and beta-lactams combined with beta-lactamase inhibitors are potential selectors of resistant strains, but fluoroquinolones (FQ) are the most concerning and dangerous antibiotics [13,14]. Recent guidelines from the Infectious Diseases Society of America (IDSA) recommend reserving their use to protect the ecosystem from MDR and harm [15].

Antimicrobial Stewardship Programs (ASPs) play a crucial role in reducing emergencies and the transmission of resistant pathogens through the advice they provide in prescription practices. The implementation of ASP actions and the data of long-term outcomes in the community are limited [16,17,18]. Previous work by our group from 2017 to 2021 showed a pronounced decrease in the incidence densities (ID) of multidrug-resistant enterobacteriaceae, such as *Escherichia coli* ESBL-producing strains, after a period of ASP intervention [19]. During this intervention, there was a marked reduction in antimicrobial consumption. The program followed a non-mandatory educational advisory model, focusing on the overall reduction in third-generation cephalosporins, amoxicillin-clavulanic acid (co-amoxclav), azithromycin, and clindamycin use, with a specific emphasis on FQ. This prompted us to investigate whether a similar trend existed for MRSA and *Clostridioides difficile*, which also indicated prescription quality.

In this study, we evaluate our hypothesis regarding the change in community-associated MRSA incidence by following the prescriptive modification of these antimicrobials through an ASP in primary care over a 5-year period.

## 2. Materials and Methods

### 2.1. Design, Setting, and Study Periods

This quasi-experimental before-and-after comparison study was conducted in the Lleida region, which is part of the public healthcare network of Catalonia (CatSalut), Spain, during the period from January 2014 to December 2021, with an interventionist approach starting in January 2017 (5 years). The general practitioners and pediatricians in the region served a reference population of 340,000 inhabitants across 23 primary care centers in direct coordination with a regional microbiology laboratory and a level III referral hospital.

In 2016, the Infection and Antibiotic Policy Territorial Commission, consisting of professionals from various specialties, groups, and administrations, launched a specific ASP for the community [20], as part of a larger regional translational program (P-ILEHRDA) that already encompassed other settings such as acute hospitals, long-term care facilities, and geriatric residences. The program design was based on the consensus document on ASPs published by the Spanish Society of Infectious Diseases and Clinical Microbiology (SEIMC), adapted to the territorial characteristics [21]. Administrative recognition from the management was obtained for its implementation.

The ASP implementation relied on interdisciplinary and multidisciplinary actions from professionals, including operational teams in each primary care center composed of at least a general practitioner, a nurse, and a pediatrician. Additionally, a coordinating technical team included general practitioners, hospital infectious disease specialists, pediatricians, microbiologists, primary care pharmacists, community pharmacists, geriatricians, emergency physicians, podiatrists, and dentists. The clinical references were selected based on their interest, knowledge, experience, analytical skills, relationship with the teams, and proficiency in providing training.

The program encompassed the following educational and training actions: (1) Periodic development and updating of regional diagnostic and antibiotic treatment protocols for the most prevalent infections (urinary tract, respiratory, skin and soft tissue, and odontogenic infections), based on scientific evidence; (2) the creation of a free-download APP (ProAPP Lleida) for access to this documentation, which was also available on the institution’s intranet; (3) regular general and specific structured training, both in-person and online, for professionals through the courses, sessions, workshops, or seminars; (4) daily review by operational teams of all the positive microbiological results from the centers and weekly review of prescriptions for the study’s specific antibiotics, except on weekends and holidays; (5) daily non-mandatory virtual written educational advice on computerized SAP “Systems, Applications, Products in Data Processing” or E-cap “Primary Care Clinical Station” and direct personalized advice, in-person or by telephone, to prescribing medical professionals. The advice emphasized the appropriateness of empirical treatments, treatments tailored to microbiological results, dose adjustments, therapeutic de-escalation, shortened duration of treatment, presence of toxicity, or interactions; (6) preparation of monitoring reports on consumption, incidence density of multidrug-resistant microorganisms, and local microbiological sensitivity for annual comparative evaluation between the centers and feedback to the teams. The actions were not contingent on extra remuneration for professionals. The work and action diagram has been described in previous publications [19].

No restrictive prescription measures were implemented in any of the study periods. The typology of recommendations was prospectively collected to assess the incidence over time. The advisories were only discontinued in 2020 due to the onset of the SARS-CoV-2 pandemic.

### 2.2. Sources of Information

The information on community prescription and microbiological resistance was obtained from the regional dispensing data and the databases of an integrated departmental management program, respectively. For the temporal analysis, the updated semiannual number of inhabitants with a health card was collected.

### 2.3. Measurement of Consumption and Microbiological Impact Outcomes

The primary outcome of the study was the change in the overall consumption of antimicrobials in the community, specifically non-recommended antimicrobials (NRA), due to a higher risk of resistance or a high spectrum index (HSI). These included FQ, cephalosporins, co-amoxiclav, clindamycin, and azithromycin, which were analyzed every semester during the intervention period from 2017 to 2021 and compared to a previous reference period.

The secondary outcome focused on the trend in the evolution of *S. aureus*, both methicillin-sensitive (MSSA) and methicillin-resistant (MRSA), and their resistance to levofloxacin, clindamycin, and erythromycin.

A third input considered in the study was the presence of cases of pathological diarrhea caused by *C. difficile* in the community, whether requiring hospital admission or not; this was attributed to the outcomes of the ASP.

### 2.4. Evaluation Methods

The calculation of antimicrobial pharmaceutical consumption utilized the methodology of the Anatomical Therapeutic Chemical Classification System and Defined Daily Doses (ATC/DDDs) established by the World Health Organization (WHO), which was revised in 2023 (http://www.whocc.no/atc_ddd_index/) (accessed on 26 September 2023). The consumption was expressed as the number per 1000 inhabitants per day over the study population with a health card (DID). Defined Daily Doses (DDDs) represent the average maintenance doses per day for the antibiotic used in its first indication.

The evolutionary impact on resistances was assessed by calculating the ID per 1000 inhabitants per day for the mentioned microorganisms, semiannually; this assessment was similar to that for the antimicrobial consumption. Only one culture per person and semester was considered for the calculation. It was assumed that there would be a 6-month delay between the intervention, implementation, and any associated changes in resistance, as suggested in some articles [22]. Therefore, the temporal analysis of resistances extended for an additional 6 months beyond the study period. The resistance percentage was identified as resistant samples among the total antibiograms performed. The standard international criteria proposed by Magiorakos et al. [23] were used for defining bacterial multidrug resistance. The identification of new cases, based on a single clinical sample, was provided by the Regional Microbiology Section, which determined antibiotic resistance by following the recommendations of the European Committee on Antimicrobial Susceptibility Testing (EUCAST) [24].

While *C. difficile* is not considered an MDR, it is included in national and European surveillance due to its clinical–epidemiological significance. The definitions recently issued by the European Society of Clinical Microbiology and Infectious Diseases [25] were used for the calculations and case identification.

### 2.5. Statistical Analysis

The continuous quantitative variables were expressed as mean ± standard deviation (SD), while the categorical quantitative variables were presented as frequencies and percentages (%). The graphical representations of the antibiotic consumption and resistance evolution were created using line histograms, highlighting the cut-off point between the pre- and post-intervention periods. The main measure of association used was the relative risk (RR) or relative change between incidence densities. For the resistance measured in the rates, the odds ratio (OR) was employed. To assess the impact in absolute terms (in ID), attributable risk (absolute effect) was used, and in relative terms, the preventable fraction (relative effect) in the intervention was used and was expressed as a percentage. The analysis of the attributable effects of intervention was calculated by comparing the pre- and post-intervention periods at three cut-off points: the beginning, the middle, and the end of the intervention period. The temporal trend in each period, pre- and post-intervention, was analyzed using the chi-square test. Changes in quantitative variables such as ID were analyzed using the Student–Fisher *t*-test and one-way ANOVA. All the estimates were accompanied by the corresponding 95% confidence interval (CI). The accepted confidence level was *p* < 0.05, and the statistical package used was EPIDAT (version 3.1) from the Pan American Health Organization.

## 3. Results

During the study period (2014 to 2021), a total of 11,814,508 DDDs of oral antimicrobials were dispensed; they were prescribed by 349 primary care consultants (312 general practitioners and 37 pediatricians) in the Lleida health region. The average semiannual post-intervention population consisted of 342,086 inhabitants, compared to 335,046 inhabitants in the pre-intervention period (2014 to 2016).

Between 2017 and 2021, a total of 6856 interventions were conducted, including 1636 (23.9%) educational advisories related to positive microbiological samples for *S. aureus*; these were primarily cutaneous. There was an average annual trend of 36.6% growth in interventions, interrupted only in 2020 due to the SARS-CoV-2 pandemic. Antibiotic modification or suspension in advisory sessions was present in 1059 cases (64.7%).

### 3.1. Impact on Antibiotic Consumption

Throughout the entire study period, penicillin was the most prescribed antibiotic, accounting for 66.0% of the prescriptions. The studied NRA (co-amoxiclav, cephalosporins, FQ, azithromycin, and clindamycin) constituted 46.6% of the total antibiotics used. The temporal evolution in DID of the global prescription of any antimicrobial, including the NRA, in any period, is shown in Figure 1. The community’s overall use of antibacterials in DID decreased by 33.7% between 2017 and 2021, with the average DID between the periods experiencing a drop of −0.095 (0.325), with a standard deviation (SD) (*p* < 0.0001). Similarly, the NRA group also exhibited a significant decrease of 37.6%, declining from 1.476 (0.131) in 2014–2016 to 1.047 (0.287) in 2017–2021 (mean difference −0.432, [95% CI −0.163 to −0.701], *p* = 0.004). The semester changes in the consumption of beta-lactamase inhibitors, FQ, and cephalosporins per DID, in the pre- and post-intervention periods in the health region, are described in Figure 2.

Table 1 shows the significant reductions in the specific antibiotics used for Gram-positive infections at three points (initial, middle, and final periods) and their final impact: FQ, cephalosporins, co-amoxiclav, azithromycin, and clindamycin. Before the intervention, a significant decreasing trend in the dispensing of FQ and co-amoxiclav was observed. However, this decrease persisted in the post-intervention period, but a statistically significant early and drastic reduction occurred from the third semester of ASP implementation. The average drops in DID per semester for these ANR were −0.064 (0.173) and, specifically, −0.023 (0.137) for co-amoxiclav, −0.023 (0.030) for FQ, and −0.004 (0.014) for cephalosporins (*p* < 0.001). A similar inflection point occurred in the sixth semester for azithromycin −0.017 (0.041) and clindamycin −0.001 (0.004) (*p* < 0.001), and this decline, along with other ANR, persisted until the end of the period.

Regarding the expected trends after those observed in the previous period, the intervention was also associated with significant changes in the post-intervention prescription, with additional significant decreases of −0.2 (95% CI −0.4 to −0.1), −0.4 (−0.5 to −0.3), −0.4 (−0.5 to −0.3), −0.3 (−0.4 to −0.1) (*p* < 0.001), and −0.1 (−0.2 to −0.05) (*p* = 0.02) in DID per semester for FQ, cephalosporins, co-amoxiclav, azithromycin, and clindamycin, respectively. In contrast, the recommended antibiotics (RA) (amoxicillin, cloxacillin, and cefadroxil) did not show a proportionally relevant inverse increase.

### 3.2. Impact on Antimicrobial Resistance

The antibiotic sensitivity was tested in 3586 clinical samples of *S. aureus* collected over 8.5 years, of which 948 (26.4%) were MRSA.

There were no statistically significant variations in the methicillin resistance rates between the two periods (26.4%) (948/3586). The overall resistance rates for clindamycin, levofloxacin, and erythromycin were 22.5% (807/3586), 33.1% (1187/3586), and 31.5% (1130/3586), respectively. The proportion of *S. aureus* resistant to levofloxacin significantly decreased between the two study periods by 15.1% (371/999 to 816/2587) (OR 0.68, [95% CI 0.58 to 0.79]) (*p* < 0.001). Conversely, the resistance increased in clindamycin and erythromycin, though only significantly in the first one, with a percentage of 53.4% (123/999 to 684/2587) (OR 2.29, [95% CI 1.86 to 2.31], *p* < 0.001). Table 2 presents the semestral comparison of microbiological resistance rates according to the *S. aureus* typology. It shows statistically significant drops in levofloxacin resistance rates, in both MSSA and MRSA, in the last two sections of the intervention period (*p* = 0.035). Resistance to the studied antibiotics increased significantly in the pre-intervention period in both bacteria (*p* < 0.001). After ASP initiation and throughout the post-intervention period, the resistance rates maintained a linear trend towards a general decrease in MRSA (*p* < 0.005), particularly to FQ (OR 0.74, [95% CI 0.64 to 0.86], *p* < 0.001).

Table 3 shows the resistance changes in the IDs at three points (initial, middle, and final periods), during the intervention and in terms of the overall impact. Comparatively, there were no decreases between periods in the IDs per 1000 inhabitants and per day of *S. aureus*. However, within the intervention period, both the IDs for MRSA and those according to the studied co-resistance typology to clindamycin, levofloxacin, and erythromycin significantly decreased during the intervention period (*p* < 0.001) (Figure 3). These IDs decreased by −0.109 cases (95% CI, −0.232 to −0.064) for methicillin; −0.091 cases (−0.105 to −0.063) for clindamycin; −0.128 (−0.230 to −0.097) for levofloxacin; and −0.112 (−0.116 to −0.084) for erythromycin, with a relative reduction of 62.1%, 58.0%, 55.4%, and 62.9% (*p* < 0.001) at the end of 5 years.

The observed change in the inflection and decline occurred, with a significant linear trend, for both MRSA and its resistance phenotypes to the three studied antimicrobials, in the second semester of the intervention period (*p* < 0.001). From that moment, a modification of the slope of −0.011 cases (SD, 0.043) (*p* < 0.001) per 1000 inhabitants and day, per semester, was noted for MRSA, −0.004 cases (0.036) for clindamycin (*p* = 0.005), −0.015 cases (0.040) for levofloxacin (*p* = 0.002), and −0.010 cases (0.022) for erythromycin (*p* < 0.001).

Finally, there were 56 cases of community-acquired *C. difficile* infection. There were no instances of recurrence. The ID of the initial community-onset infection increased by 41.2% (0.004 to 0.009) during the intervention period (OR 1.24, [95% CI 0.71 to 2.18]), but this increase was not statistically significant (Figure 4).

## 4. Discussion

Our observational and quasi-experimental study suggests a long-term positive effect on community antimicrobial prescription following the implementation of an educational ASP designed for primary care. It also indicates a linear trend and an association between antibiotic use and appropriateness and the incidence density of MRSA in the community.

In recent years, the attention to antimicrobial administration has been increasing [26]. In 2023, after several years of recommendations, the European Union (EU) urged all member states to implement real action plans against antimicrobial resistance and to promote the prudent use of antibiotics [27]. However, despite a significant reduction in the average consumption of community systemic treatments in the European Economic Area (EEA) during the period of 2012–2021 (19.3 DID vs. 15.0 DID), the weighted average proportion of the EU/EEA population in relation to the consumption of penicillins, cephalosporins, FQ, and broad-spectrum macrolides (except erythromycin) compared to the narrow-spectrum ones has shown a statistically significant increasing trend of 3.7 (range of countries: 0.1–20.7) in half of the countries, including Spain [28]. These data highlight the value of our intervention, which led to a behavioral change in prescription practices, with a significant reduction in the use of levofloxacin, clindamycin, azithromycin, and cephalosporins, resulting in an increase in the use of RA, specifically first-generation cephalosporins and cotrimoxazole. However, this observed increase was not inversely proportional to these antibiotics recommended by our P-ILEHRDA program, especially in the management of skin and soft tissue infections where Gram-positive microorganisms are present. We believe that improvements in microbiological sample collection techniques, with a focus on percutaneous aspiration rather than swabs, along with the avoidance of indiscriminate culturing of chronic ulcers, which are mostly colonized and are either diagnostic confusion elements or not amenable to antimicrobial treatment, may explain this finding [29,30,31].

Multimodal models with multifaceted interventions, like those in our study, are more effective than single interventions in changing antimicrobial prescription behavior [32,33]. The studies by Arnold et al. [34] demonstrated that continuous training and feedback of results to professionals improve clinical practice in a sustained and continuous manner, supporting our case when a lower level of counseling during the COVID-19 period did not interfere with the results. Furthermore, while most studies have focused exclusively on respiratory tract infections [35,36], our ASP was designed to address all prevalent types of infections, with a comprehensive approach that was in line with expert recommendations, societies, and previous studies [21,34,37]. Restrictive interventions were not included because, although they can have rapid effects on targeted antibiotic use, such measures are negatively viewed by professionals and do not help to adjust prescription behaviors [38,39].

Our study linked antibiotic dispensing data with around 4000 positive results for *S. aureus* in various samples, mainly skin-related and routinely collected; these results provided our study with sufficient power to detect the direct relationship between consumption and resistance.

The direct association between MRSA and the use of the studied ANR has been evidenced in various analyses, depending on the volume of the exposed population and the age group [40,41,42]. The relationship has also been established at both the host and the molecular and microbiological levels. A meta-analysis of associations between individual exposures to antibiotics and the risk of MRSA acquisition showed FQ, glycopeptides, cephalosporins, macrolides, and β-lactams to be the most notable [9]. In vitro studies have shown how exposure to most of these antibiotics causes a particular co-resistance in MRSA, as opposed to MSSA, especially to ciprofloxacin, erythromycin, and clindamycin; this is perhaps due to resistance transfer and competition between pathogens [42]. Several authors [13,43,44] have demonstrated how the selectivity and the reduction in the antibiotic pressure threshold on the population can determine the molecular epidemiology of MRSA and cause different phenotypes and shifts toward more susceptible sub-lineages within all clonal complexes. In this regard, our work, although we did not perform a molecular study, not only indicated a reduction in MRSA IDs but also a significant linear downward trend in the total of its co-resistances during the ASP period.

Eliminating the pressure from selected antibiotics such as FQ may not only favor a reduction in MRSA presence but also prompt a rapid decline in resistance. Previous reviews [12,22,45,46,47] have evaluated the temporal relationship between antibiotic consumption and resistance development in outpatient and primary care settings. Bell et al. [12] included 243 studies (case-control, cross-sectional, ecological, and experimental studies) on all antibiotics and bacteria. The time between consumption and resistance was 6 months or less in 53% and more than 6 months in 23%, and it was unclear in the remaining included studies [12]. The use of FQ has been associated with MRSA incidence shortly after exposure (between 1 and 5 months) [48,49,50] and with FQ resistance between 0 and 4 months [42,51]. Similarly, the use of lincosamides was associated with the incidence of MRSA and clindamycin resistance in the second month, and in the case of penicillin + β-lactamase inhibitors, this same relationship occurred with a delay of 1 to 5 months [40,42,48,52]. However, a reversal in the trend is feasible with the same temporal intensity if suspension occurs. Studies conducted in the United Kingdom [53,54] assessed the prevalence of ciprofloxacin resistance before, during, and after a national restriction on the use of FQ [22]. These works reported a reduction in resistance levels in less than 3 months. Our trend analysis showed changes in resistance with significant inflection points in MRSA ID, highlighting a steep decline in the early semesters after the start of our ASP, followed by a sustained, significant, and intense decline until the end of the study period. This ecological effect probably occurred because the outcomes were higher at the beginning of the program, when it was easier to improve, and then were maintained over time.

Implementing outpatient interventions to reduce inappropriate antibiotic use can substantially decrease the rates of community-acquired *C. difficile* infection [55,56]. Various studies [57] and a recent meta-analysis [58] that used data from eight studies in various regions of the world have shown that exposure to various categories of antibiotics, including clindamycin, FQ, cephalosporins, penicillins, macrolides, and cotrimoxazole, was associated with an increased risk of *C. difficile* in adults. Our study, despite a significant reduction in most of these antibiotics, did not observe a decrease in *C. difficile* infection. This aspect could be explained by two reasons: the first is the availability of improved protocols and the diagnostic suspicion regarding *C. difficile* as a cause of diarrhea. Alcalá et al. [59] demonstrated in our country that it was only suspected in 47.6% of the cases. The second was the implementation in our health area of new diagnostic techniques with higher sensitivity (PCR techniques in the diagnostic algorithm) at the end of 2016.

Finally, the results of our study have other strengths. First, measuring dispensed antibiotics rather than prescribed ones is considered to be a much stronger measure of exposure and consumption since it faithfully reflects the patients who, by picking up the medication at a pharmacy, have used it. Second, having a unique central microbiology laboratory avoids changes in study techniques and biased variability in the number of samples studied. On the other hand, our work has some limitations: first, the molecular recognition of *S. aureus* ribotypes and resistance genotypes, which could help to better identify interventions aimed at avoiding antibiotics that are considered to be a high risk, were not performed. Second, the synergistic role of standard universal strategies in preventing infection, such as decolonization or hand hygiene, has not been analyzed, with the understanding that the latter has increased during the SARS-CoV-2 pandemic. Third, only community data on overall and selected antibiotic consumption were collected, while antibiotics were also prescribed in hospitals, which, in our case, already had an ASP established with a similar action methodology, which could have magnified the results on the reduction in resistance.

## 5. Conclusions

The results of this study demonstrate that after five years, the implementation strategies of an educational community ASP, aimed at reducing antibiotic pressure, were associated with significant benefits in terms of both antimicrobial consumption and the local ecological impact of MRSA.

## Figures and Tables

**Figure 1 antibiotics-13-00092-f001:**
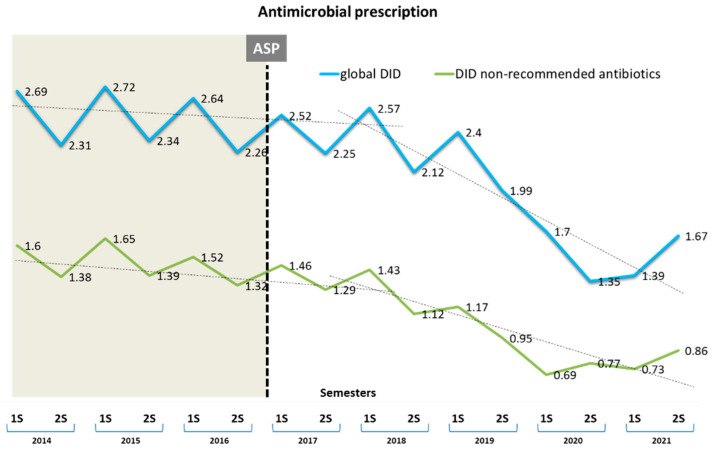
Semestral evolution of defined daily doses per 1000 inhabitants/day (DID). Global antimicrobial prescription (upper line) and non-recommended antimicrobials (NRA) (lower line). Shaded area represents the pre-antibiotic stewardship program (ASP) intervention period.

**Figure 2 antibiotics-13-00092-f002:**
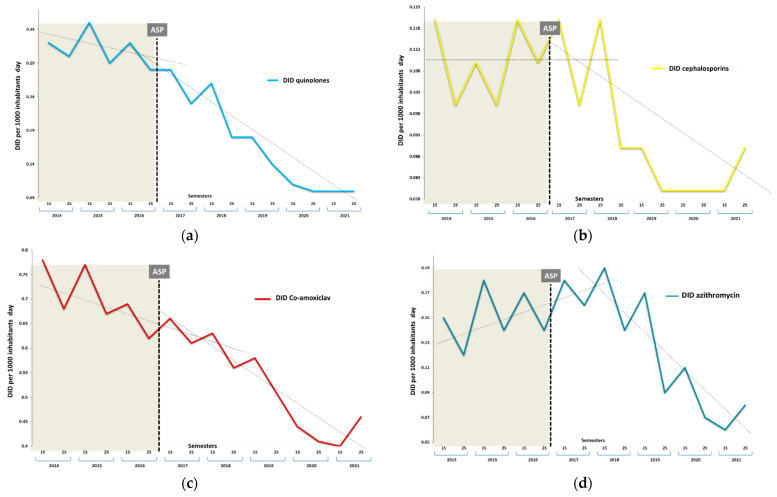
Antimicrobial prescription of other antimicrobials. Semestral evolution of defined daily doses per 1000 inhabitants/day (DID). (**a**) Quinolones, (**b**) cephalosporins, (**c**) co-amoxclav, (**d**) azithromycin, (**e**) clindamycin, (**f**) recommended antimicrobials—amoxicillin, cloxacillin, cefadroxil, cotrimoxazole. 1S; first semester, 2S; second semester.

**Figure 3 antibiotics-13-00092-f003:**
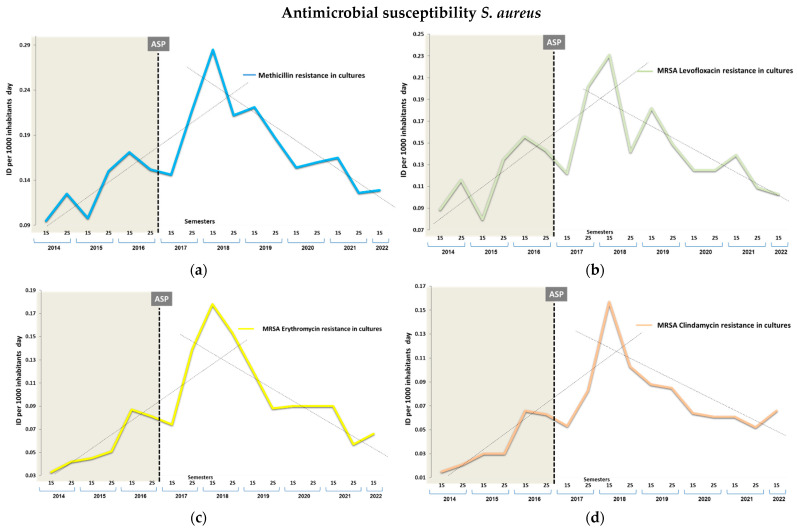
Semestral evolution MRSA (**a**) and antibiotic-resistant MRSA in general cultures per incidence density (ID) per 1000 inhabitants/day. (**b**) Levofloxacin resistance, (**c**) erythromycin resistance, (**d**) clindamycin resistance. 1S; first semester, 2S; second semester.

**Figure 4 antibiotics-13-00092-f004:**
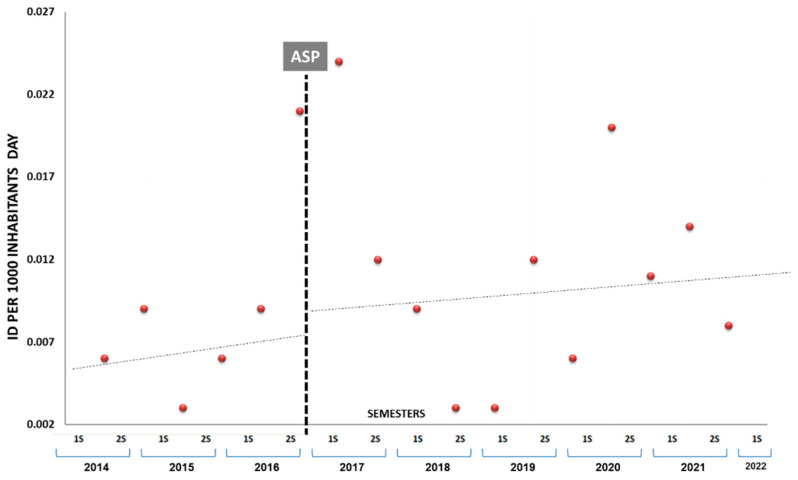
*C difficile* infection incidence density (ID) per 1000 inhabitants/day over study period.

**Table 1 antibiotics-13-00092-t001:** Changes in antimicrobial prescription (ATC codes J01 and specific antimicrobials) before and after ASP intervention at 3 points (initial, middle, and final periods) and overall impact.

Prescribed Antibiotic	DID Pre-Intervention Period	Relative Change First Semester 2017 (95% CI)	Relative Change First Semester 2019 (95% CI)	Relative Change Second Semester 2021 (95% CI)	Absolute Effect Post-Intervention Period	Relative Effect (%)
Total antibiotics (J01)	2.496	0.892 (0.890 to 0.894)	0.790 (0.787 to 0.781)	0.670 (0.668 to 0.672)	−0.688 (−0.691 to −0.685)	27.57 (27.65 to 27.49)
Total antibiotics not recommended (ANR)	1.476	0.989 (0.987 to 0.992)	0.796 (0.793 to 0.797)	0.581 (0.579 to 0.583)	−0.079 (−0.079 to −0.079)	37.57 (37.48 to 37.66)
Co-amoxclav (J01CR02)	0.704	0.940 (0.938 to 0.943)	0.821 (0.819 to 0.824)	0.659 (0.657 to 0.662)	−0.250 (−0.251 to −0.249)	35.59 (35.50 to 35.68)
Quinolones (J01M)	0.311	0.903 (0.897 to 0.908)	0.588 (0.584 to 0.593)	0.328 (0.325 to 0.331)	−0.294 (−0.295 to −0.294)	94.74 (94.72 to 94.75)
Ciprofloxacin (J01MA02)	0.114	0.779 (0.770 to 0.788)	0.556 (0.549 to 0.564)	0.439 (0.433 to 0.446)	−0.052 (−0.052 to −0.051)	45.39 (45.07 to 45.70)
Levofloxacin (J01MA12)	0.132	1.055 (1.046 to 1.065)	0.730 (0.722 to 0.738)	0.338 (0.332 to 0.344)	−0.065 (−0.065 to 0.064)	49.18 (48.93 to 49.44)
Cephalosporins (J01D)	0.111	1.115 (1.104 to 1.126)	0.785 (0.776 to 0.794)	0.807 (0.798 to 0.816)	−0.025 (−0.026 to −0.025)	22.99 (22.61 to 23.38)
Cefuroxime (J01DC02)	0.061	0.739 (0.726 to 0.751)	0.614 (0.603 to 0.625)	0.433 (0.424 to 0.442)	−0.025 (−0.025 to −0.025)	40.96 (40.50 to 41.41)
Third-generation cephalosporins (J01DD)	0.046	1.223 (1.204 to 1.242)	0.967 (0.971 to 1.004)	1.275 (1.256 to 1.294)	0.001 (0.001 to 0.001)	2.32 (1.57 to 3.07)
Azithromycin (J01FA10)	0.152	1.204 (1.194 to 1.213)	1.119 (1.110 to 1.128)	0.533 (0.527 to 0.539)	−0.042 (−0.043 to −0.041)	27.67 (27.36 to 27.98)
Clindamycin (J01FF01)	0.021	0.771 (0.750 to 0.793)	0.720 (0.699 to 0.741)	0.846 (0.824 to 0.869)	−0.004 (−0.005 to −0.004)	21.57 (20.61 to 22.51)
Total recommended antibiotics (RA)	0.969	1.032 (1.028 to 1.035)	1.146 (1.143 to 1.150)	0.748 (0.746 to 0.751)	−0.052 (−0.052 to −0.051)	21.29 (21.17 to 22.42)
Amoxicillin (J01CA04)	0.925	1.028 (1.027 to 1.029)	1.081 (1.081 to 1.082)	0.711 (0.709 to 0.712)	−0.218 (−0.218 to −0.217)	23.53 (23.48 to 23.59)
Cloxacillin (J01CF02)	0.018	1.018 (0.991 to 1.046)	1.032 (1.005 to 1.060)	0.722 (0.700 to 0.745)	−0.004 (−0.005 to −0.004)	25.17 (24.18 to 26.15)
Cefadroxil (J01DB05)	0.001	1.652 (1.418 to 1.924)	2.702 (2.380 to 3.066)	8.835 (8.066 to 9.678)	0.002 (0.002 to 0.002)	84.02 (82.80 to 85.14)
Cotrimoxazole (J01EE01)	0.027	1.158 (1.134 to 1.182)	1.416 (1.390 to 1.443)	1.956 (1.924 to 1.988)	0.014 (0.013 to 0.014)	33.89 (33.26 to 34.52)

ATC; Anatomical Therapeutic Chemical. DID; defined daily doses per 1000 inhabitants per day. NRA; non-recommended antibiotics (co-amoxiclav, quinolones, cephalosporins, azithromycin, and clindamycin). RA; recommended antibiotics (amoxicillin, cloxacillin, cefadroxil, and cotrimoxazole).

**Table 2 antibiotics-13-00092-t002:** Rates of microbiological resistance of *S. aureus*.

	Antimicrobial Resistance	Comparisons by Semesters (S)													
		Second S 2016 vs. Second S 2017				Second S 2016 vs. Second S 2019				Second S 2016 vs. First S 2022					
		% (n/N) Pre-Intervention Resistance	% (n/N) Post-Intervention Resistance	OR CI 95%	*p*	% (n/N) Pre-Intervention Resistance	% (n/N) Post-Intervention Resistance	OR CI 95%	*p*	Prevention Rate (%)	% (n/N) Pre-Intervention Resistance	% (n/N) Post-Intervention Resistance	OR CI 95%	*p*	Prevention Rate (%)
MSSA	Clindamycin	6.54 (48/734)	16.08 (23/143)	2.73 (1.60–4.67)	<0.001	6.53 (48/734)	17.89 (34/190)	3.11 (1.94–4.99)	<0.001	NA	6.53 (48/734)	19.10 (34/178)	3.37 (2.09–5.42)	<0.001	NA
	Levofloxacin	17.71 (130/734)	13.22 (19/143)	0.71 (0.42–1.19)	NS	17.71 (130/734)	6.84 (13/190)	0.34 (0.18–0.61)	<0.001	61.4 (33.2–77.6)	17.71 (130/734)	10.11 (18/178)	0.52 (0.30–0.88)	0.014	42.9 (9.10–64.1)
	Erythromycin	19.20 (141/734)	22.07 (33/143)	1.26 (0.82–1.93)	NS	19.20 (141/734)	21.57 (41/190)	1.15 (0.78–1.71)	NS	NA	19.20 (141/734)	22.47 (40/178)	1.21 (0.81–1.81)	NS	NA
MRSA	Clindamycin	28.30 (75/265)	38.35 (28/73)	1.57 (0.91–2.71)	NS	28.30 (75/265)	45.31 (29/64)	2.09 (1.19–3.67)	0.009	NA	28.30 (75/265)	51.11 (23/45)	2.64 (1.39–5.03)	0.002	NA
	Levofloxacin	90.94 (241/265)	93.15 (68/73)	1.35 (0.49–3.68)	NS	90.94 (241/265)	79.63 (50/64)	0.39 (0.18–0.81)	0.010	12.37 (0.27–23.01)	90.94 (241/265)	81.25 (39/45)	0.43 (0.18–0.99)	0.044	10.6 (−2.83–22.4)
	Erythromycin	42.64 (113/265)	56.62 (47/73)	1.75 (1.06–2.88)	0.026	42.64 (113/265)	46.87 (30/64)	1.18 (0.68–2.05)	NS	NA	42.64 (113/265)	51.11 (23/45)	1.40 (0.74–2.64)	NS	NA

(n/N); n; total positive antibiograms, N; total antibiograms, MSSA; methicillin-sensitive *staphylococcus aureus*. MRSA; methicillin-resistant *staphylococcus aureus.* NS; not significant. NA; not applicable; OR: qdds ratio.

**Table 3 antibiotics-13-00092-t003:** Changes on incidence density before and after ASP intervention at 3 points (initial, middle, and final periods) and overall impact.

Antimicrobial Resistance	ID Pre-Intervention Period (95% CI)	Relative Change Second Semester 2017 (95% CI)	Relative Change Second Semester 2019 (95% CI)	Relative Change First Semester 2022 (95% CI)	Absolute Effect Post-Intervention Period	Relative Preventable Effect (%)
MSSA						
Clindamycin	0.024 (0.024 to 0.024)	2.860 (1.740 to 4.703)	4.170 (2.687 to 6.471)	4.071 (2.623 to 6.371)	0.065 (0.054 to 0.076)	73.11 (63.67 to 80.10)
Levofloxacin	0.065 (0.064 to 0.065)	0.872 (0.539 to 1.412)	0.588 (0.332 to 1.041)	0.795 (0.486 to 1.302)	−0.008 (−0.021 to 0.005)	12.33 (−8.65 to 29.25)
Erithromycin	0.071 (0.070 to 0.071)	1.397 (0.956 to 2.040)	1.712 (1.209 to 2.424)	1.630 (1.147 to 2.316)	0.041 (0.026 to 0.056)	39.92 (23.81 to 47.77)
MRSA						
Clindamycin	0.037 (0.037 to 0.038)	2.229 (1.444 to 3.440)	2.276 (1.483 to 3.494)	1.762 (1.104 to 2.812)	0.040 (0.028 to 0.052)	51.93 (38.18 to 62.62)
Levofloxacin	0.120 (0.119 to 0.120)	1.684 (1.287 to 2.204)	1.246 (0.921 to 1.685)	0.858 (0.604 to 1.218)	0.022 (0.003 to 0.041)	15.48 (1.80 to 27.26)
Erithromycin	0.056 (0.056 to 0.057)	2.483 (1.767 to 3.489)	1.563 (1.045 to 2.337)	1.169 (0.747 to 1.831)	0.045 (0.031 to 0.060)	44.73 (31.96 to 55.10)

ID; incidence density per 1000 inhabitants per day. MSSA; methicillin-sensitive *Staphylococcus aureus*. MRSA; methicillin-resistant *Staphylococcus aureus.*

## Data Availability

No new data were created or analyzed in this study. Data sharing is not applicable to this article.

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
