# Peer review of "Effects of a Primary Care Antimicrobial Stewardship Program on Meticillin-Resistant Staphylococcus aureus Strains across a Region of Catalunya (Spain) over 5 Years"

_antibiotics, 2024, doi:10.3390/antibiotics13010092_

Round 1

Reviewer 1 Report

Comments and Suggestions for Authors

Comments to the authors  

1.      How was the feasibility and effectiveness of the prolonged educational counseling ASP assessed in the large regional outpatient setting?

2.      Were there any specific challenges encountered during the implementation of the ASP, and if so, how were they addressed?

3.      Can you elaborate on the trends in antibiotic prescriptions, especially focusing on the reduction in harmful effects and resistance selection?

4.      What were the most commonly prescribed antibiotics, and how did their usage change over the study period?

5.      How did the community's overall use of antibacterials change, and were there specific antibiotics that showed a significant decrease?

6.      Can you discuss the significant reductions in specific antibiotics used for Gram-positive infections, as presented in Table 1?

7.      What factors might have contributed to the observed early and drastic reduction in certain antibiotics, such as fluoroquinolones and co-amoxiclav?

8.      What were the overall antibiotic resistance rates for methicillin, clindamycin, levofloxacin, and erythromycin in Staphylococcus aureus?

9.      How did the proportion of Staphylococcus aureus resistant to levofloxacin change between the two study periods, and what might explain this change?

10.  Can you elaborate on the changes in resistance at different points during the intervention and the overall impact, as presented in Table 3?

11.  Were there specific co-resistance patterns that showed a significant decrease during the intervention period?

12.  How effective were the advisory sessions, and what proportion of cases resulted in antibiotic modification or suspension?

13.  Were there specific types of microbiological samples for which educational advisories were more common?

14.  What implications do the observed changes in antibiotic consumption and resistance have on public health, especially considering the regional scale of the study?

15.  The authors are advised to Improve English in the text using more scientific rather than descriptive language. 

Comments on the Quality of English Language

The authors are advised to improve the English in the text by incorporating more scientific language rather than relying on descriptive language.

Author Response

We sincerely appreciate the overall positive evaluation of our article. We will now proceed to respond to the questions posed by the reviewer, as we understand they aim to provide a better understanding of the program and its results.

  1. How was the feasibility and effectiveness of the prolonged educational counseling ASP assessed in the large regional outpatient setting?

The viability and efficacy of the ASP were assessed through the execution and analysis of indicators related to the use and consumption of antibiotics, as established by the Spanish National Plan against Antibiotic Resistance (PRAN) https://www.resistenciaantibioticos.es/es/lineas-de-accion/control/programas-de-optimizacion-de-uso-de-los-antibioticos-proa

  1. Were there any specific challenges encountered during the implementation of the ASP, and if so, how were they addressed?

The main challenges included a lack of specific funding, which was addressed through the motivation and interest of professionals, non-healthcare private entities, and local healthcare managers. Secondly, the initial period (2020-21) of the SARSCoV2 pandemic required emotional reinforcement in engaging professionals due to workload overload. This period also required the maintenance of training and consultancy through an online communication system.

  1. Can you elaborate on the trends in antibiotic prescriptions, especially focusing on the reduction in harmful effects and resistance selection?

After the preliminary results previously published by our group regarding the reduction in consumption and resistances, at least in enterobacteria, our next objective is to analyze and describe, in an upcoming article, the incidence of side effects, primarily caused by fluoroquinolones. At this time, we cannot provide you with this information.

  1. What were the most commonly prescribed antibiotics, and how did their usage change over the study period?

The two most frequently prescribed antibiotics were amoxicillin and amoxicillin-clavulanic acid, respectively. Table 1 displays the change and reduction in usage of each separately during the study period.

  1. How did the community's overall use of antibacterials change, and were there specific antibiotics that showed a significant decrease?

As shown in Table 1, the overall use of antimicrobials was modified by 27.57%, decreasing from defined daily doses (DDD) in the period 2014-16 of 2493 (SD, 0.153) to 1808 (SD, 0.208) in the post-intervention period 2017-2021. Regarding specific antibiotics, the majority experienced a significant decline as described in Table 1, except for third-generation cephalosporins, cefadroxil, and cotrimoxazole.

  1. Can you discuss the significant reductions in specific antibiotics used for Gram-positive infections, as presented in Table 1?

As stated in the previous section, almost all specific antimicrobials used for Gram-positive infections, and those not recommended, significantly decreased (p<0.001). Similarly, this trend was observed with some of the recommended antimicrobials, such as amoxicillin and cloxacillin. In the case of the latter two, possibly indicative of unnecessary prescription, as discussed.

  1. What factors might have contributed to the observed early and drastic reduction in certain antibiotics, such as fluoroquinolones and co-amoxiclav?

One of the most significant initial general and crosscutting actions was the avoidance of quinolone use, regardless of the reason and infectious entity, in favor of alternatives if available. In the case of co-amoxiclav, the substitution at the program's outset of this antibiotic with phenoxymethylpenicillin for managing pyogenic tonsillitis and cefadroxil for cellulitis treatment primarily caused in the community by methicillin-sensitive Streptococcus and S. aureus.

  1. What were the overall antibiotic resistance rates for methicillin, clindamycin, levofloxacin, and erythromycin in Staphylococcus aureus?

This aspect is specified in the manuscript in section 3.2. “There were no statistically significant variations in methicillin resistance rates between the two periods (26.4%). The overall resistance rates for clindamycin, levofloxacin, and erythromycin were 22.5%, 33.1%, and 31.5%, respectively”

  1. How did the proportion of Staphylococcus aureus resistant to levofloxacin change between the two study periods, and what might explain this change?

In section 3.2, it is presented that “The proportion of S. aureus resistant to levofloxacin significantly decreased between the two study periods by 15.1% (OR 0.68, [95% CI 0.58 to 0.79]) (p<0.001)”. The main reason that can explain this is the drastic and differentiated reduction in the total prescription of quinolones compared to other antimicrobials (94.7%) - see Table 1-.

  1. Can you elaborate on the changes in resistance at different points during the intervention and the overall impact, as presented in Table 3?

As mentioned in the text, no changes were observed in the incidence density per 1000 inhabitants per day between the two periods for S. aureus, both overall and in its variants sensitive to methicillin (MSSA) or resistant (MRSA), nor in their resistances in both to clindamycin, levofloxacin, and erythromycin. However, in the case of MRSA and its co-resistances, a statistically significant reduction in incidence densities occurred only during the intervention period, as specified in the text. “These ID decreased by -0.109 cases (95% CI, -0.232 to -0.064) for methicillin, -0.091 cases (-0.105 to -0.063) for clindamycin, -0.128 (-0.230 to -0.097) for levofloxacin, and -0.112 (-0.116 to -0.084) for erythromycin, with a relative reduction of 62.1%, 58.0%, 55.4%, and 62.9% (p<0.001) at the end of 5 years”.

  1. Were there specific co-resistance patterns that showed a significant decrease during the intervention period?

The response to this is addressed in question number 10.

  1. How effective were the advisory sessions, and what proportion of cases resulted in antibiotic modification or suspension?

The acceptance of counseling was 89.3% out of the 1636 consultations conducted, with a proportion of immediate modification or suspension at 64.7% (1059), as specified in section 3 of the text.

  1. Were there specific types of microbiological samples for which educational advisories were more common?

This aspect is particularly evident in the case of skin samples when the collection did not correspond to percutaneous aspiration or biopsy, as discussed.

  1. What implications do the observed changes in antibiotic consumption and resistance have on public health, especially considering the regional scale of the study?

This aspect has not yet been evaluated, but we believe that it is evident at least concerning mortality. Once we can verify the incidence of side effects, as discussed in question 3, we will conduct the relevant study to confirm the suspicion and our desire.

  1. The authors are advised to Improve English in the text using more scientific rather than descriptive language.

We appreciate the recommendation, and we have reviewed the English in the text. However, we believe that, due to the nature of the work, it lends itself to a scientific description throughout.

Reviewer 2 Report

Comments and Suggestions for Authors

The authors present a longitudinal analysis of the effects of the antimicrobial stewardship program on Staphylococcus aureus strains collected from the Spanish region of Catalunya. This work is motivated and built upon similar previous studies of this author group and is now focussed on antimicrobial resistance patterns in MRSA. Taking into account the global efforts for regulating the use of antibiotics, the associated healthcare costs, and understanding the utilities and responsibilities of antimicrobial stewardship programs, studies like this presented provide an important context on what are we gaining or losing with the contemporary efforts and what could be done to make the situation better.

Major issues/requested changes:

1. The figures and tables should be self-explanatory. The current captions and legends are not duly descriptive and/or sufficient to understand the presented data properly. For instance, Table 1 denotes each antibiotic in a composite row set-up. It is unclear what the name in parentheses under an antibiotic category means/refers to. Similarly, the quoted ranges are not defined and are, hence, difficult to interpret.

2. The authors resort to the use of percentages to denote the increase or decrease in quantities before or after the stewardship program intervention. Although it is helpful to draw inferences from these percent values, it is necessary that the authors provide the specific values (counts) for these associated percent. This could be provided in the form of a supplementary information table that is appropriately referenced.

3. It is desirable to understand the specific genomic/genetic modifications observed in the collected MRSA (and non-MRSA) strains over the longitudinal course of this study. Even the whole-genome analysis of a small subset of strains from different time points will enormously aid in improving the scientific contributions of this work. The authors can demonstrate the changes in the resistome and mobilome recorded over this period and put that in context with the influences from the antimicrobial stewardship program.

4. The ESKAPE pathogens, including MRSA, are known to have acute nosocomial infections. It would be helpful to have the authors dissect their dataset or results into strains from immuno-compromised patients or from intensive care regions of hospitals and clinical facilities and those collected from other sources. This will help in gaining further insights into how the MDR/XDR/PDR pathogens evolve with changes to the routine use of antimicrobials in non-nosocomial settings and how should the ASPs attempt to address these scenarios.

Comments on the Quality of English Language

The English language quality is fairly good. There are a few typographical errors and some instances of convoluted sentence phrasing which could be simplified for ease of reading.

Along a similar note, it is unclear why page 10 is left blank, and the manuscript text continues abruptly after a few figures and multiple pages. If it helps, the authors can move all page-long figures to the end to enhance the reading experience.

Author Response

We sincerely appreciate the overall positive evaluation of our article. We will now proceed to respond to the questions posed by the reviewer, as we understand they aim to provide a better understanding of the program and its results.

  1. The figures and tables should be self-explanatory. The current captions and legends are not duly descriptive and/or sufficient to understand the presented data properly. For instance, Table 1 denotes each antibiotic in a composite row set-up. It is unclear what the name in parentheses under an antibiotic category means/refers to. Similarly, the quoted ranges are not defined and are, hence, difficult to interpret.

Some table titles have been modified, and definitions of the referenced antibiotic categories have been added within parentheses. Additionally, ranges have been defined to enhance interpretation.

  1. The authors resort to the use of percentages to denote the increase or decrease in quantities before or after the stewardship program intervention. Although it is helpful to draw inferences from these percent values, it is necessary that the authors provide the specific values (counts) for these associated percent. This could be provided in the form of a supplementary information table that is appropriately referenced.

Counts have been added to the percentages presented in the text in section 3.2 when referring to comparative quantities before and after, thus avoiding the creation of new tables.

  1. It is desirable to understand the specific genomic/genetic modifications observed in the collected MRSA (and non-MRSA) strains over the longitudinal course of this study. Even the whole-genome analysis of a small subset of strains from different time points will enormously aid in improving the scientific contributions of this work. The authors can demonstrate the changes in the resistome and mobilome recorded over this period and put that in context with the influences from the antimicrobial stewardship program.

The information obtained from the analysis of the complete genome would undoubtedly be of great interest. Unfortunately, it is not the objective of our current study. Nevertheless, we are considering conducting it in an upcoming work, coinciding with a comprehensive evolutionary analysis of the resistome throughout the entire region (hospital, primary care, socio-health units, and geriatric residences) during this period.

  1. The ESKAPE pathogens, including MRSA, are known to have acute nosocomial infections. It would be helpful to have the authors dissect their dataset or results into strains from immuno-compromised patients or from intensive care regions of hospitals and clinical facilities and those collected from other sources. This will help in gaining further insights into how the MDR/XDR/PDR pathogens evolve with changes to the routine use of antimicrobials in non-nosocomial settings and how should the ASPs attempt to address these scenarios.

This aspect has been analyzed in previous works by the group, limited to the hospital setting, after the implementation of an Antimicrobial Stewardship Program (ASP) between 2013 and 2017, focusing on changes in antibiotic consumption and typology. Once again, this suggestion is not the objective of our current study. However, we hope to describe it in an upcoming work, as mentioned in the previous point, after completing the comprehensive implementation of the PROA across the entire region, encompassing hospitals, primary care, socio-health units, and geriatric residences.   

  1. Finally, following the reviewer's suggestions, Figures 3 and 4 have been moved to the end of section 3 to facilitate the reading of the text..

Round 2

Reviewer 2 Report

Comments and Suggestions for Authors

I appreciate the authors' efforts in incorporating my feedback into their revised manuscript. I am fairly satisfied with this updated version of the manuscript.

Although a few of the points raised in my previous review have not been addressed here, I understand that the authors consider those questions to be a better fit for a separate exploratory study.